# Effect of Marinating in Dairy-Fermented Products and Sous-Vide Cooking on the Protein Profile and Sensory Quality of Pork Longissimus Muscle

**DOI:** 10.3390/foods12173257

**Published:** 2023-08-30

**Authors:** Agnieszka Latoch, Małgorzata Moczkowska-Wyrwisz, Piotr Sałek, Ewa Czarniecka-Skubina

**Affiliations:** 1Department of Animal Food Technology, University of Life Sciences in Lublin, 20-950 Lublin, Poland; 2Department of Food Gastronomy and Food Hygiene, Institute of Human Nutrition Sciences, Warsaw University of Life Sciences (WULS), 02-787 Warsaw, Poland; malgorzata_moczkowska@sggw.edu.pl (M.M.-W.); piotr_salek@sggw.edu.pl (P.S.); ewa_czarniecka-skubina@sggw.edu.pl (E.C.-S.)

**Keywords:** sous-vide, marinating, proteins, sensory properties, kefir, yogurt, buttermilk

## Abstract

The aim of the study was to evaluate the effect of marinating (3 or 6 days) in kefir (KE), yogurt (YO) and buttermilk (BM) and sous-vide cooking (SV) at 60 or 80 °C on changes in the protein profile of pork in relation to its sensory quality. In the marinated raw meat, an increased share of some fractions of myofibrillar and cytoskeletal proteins and calpains were found. The greatest degradation of proteins, regardless of time, was caused by marinating in YO and KE and cooking SV at 80 °C. The lowest processing losses were in samples marinated in KE and YO and cooked SV at 60 °C, with marinating time having no significant effect. The odor, flavor, tenderness and juiciness of meat marinated in BM was better than in KE and YO. Meat marinated and cooked SV at 60 °C was rated better by the panelists. Changes in proteins significantly affect the formation of meat texture, tenderness and juiciness, which confirms the correlations. This is also reflected in the sensory evaluation. During the process of marinating and cooking meat, protein degradation should be taken into account, which can be a good tool for shaping the sensory quality of cooked pork.

## 1. Introduction

Clean label food that is natural, fresh and free of synthetic additives and preservatives is gradually gaining consumer interest. Therefore, in meat processing and scientific research, additives and preservatives are limited in favor of natural ones [1,2], which can be used in the form of marinades. Marinades are liquids containing various ingredients such as vinegar, wine, fermented dairy products, sour fruit and fruit juices, sugar, spices, herbs, salt, oil, phosphates, acids, aroma components and other additives by using different techniques (Table 1). Marination has been applied to meat and meat products to improve sensory attributes (color, tenderness, juiciness, flavor, palatability) and microbial quality [3].

Fermented dairy products (FDP) such as kefir, yogurt and buttermilk are also used as marinades. FDPs have a low pH (4.6) and contain live cultures of microorganisms, mainly lactic acid bacteria (LAB). These microorganisms, in addition to health benefits, have an antimicrobial effect on pathogens due to the production of organic acids, mainly lactic acid and its metabolites, especially bacteriocins. The addition of LAB improves sensory qualities (color, appearance, taste and acceptability) [19,20] and the microbiological quality of stored marinated meat [8,10,14,21].

The various ingredients of marinades, apart from preserving or marinating, are mainly used for the improvement of the textural properties of the meat, like tenderness and juiciness [22,23] and flavor enhancement [24]. Natural antioxidants from plants have been used to extend the shelf life of meat, among others, by reducing lipid oxidation [25].

Among the various methods of extending the shelf life of meat, the sous-vide method has gained popularity in recent years. Sous-vide (SV) cooking is a method in which food is vacuum-packed and cooked under strictly controlled time and temperature. A typical SV in meat processing is performed at 50–85 °C for several hours, depending on the intramuscular connective tissue, myofibrillar protein components, thickness, and type of meat [26,27,28]. This method effectively increases the yield of the process by limiting thermal leakage and, at the same time, positively affects the nutritional value and sensory quality [28,29,30,31]. The advantages of cooking meat in the sous-vide method are the higher nutritional value due to the higher concentration of nutrients [32], such as vitamins and the water-insoluble unsaturated fatty acids [33]. Sous-vide has a positive effect on sensory characteristics such as tenderness, juiciness and color [28,29,30,31,32,34,35,36] at higher temperatures (75 °C, 100 °C) to better preserve the volatile profile of meat, avoiding the accumulation of off-flavor such as hexanal or 3-octanone in comparison to traditional cooking methods [32,37] as well as mask the negative sensory features of meat for example aroma and taste of pig boar [38]. All these features are strictly dependent on the protein content, their composition and their chemical state.

In cooked meat, tenderness is the main criterion for quality assessment, which is associated with changes in muscle tissue, including intramuscular connective tissue and myofibrillar protein components [30,38]. According to some authors [32,39], opposite to traditional methods, low-temperature cooking of SV creates overpressure due to saturated steam. This phenomenon helps to maintain the overall cellular structure by minimizing interactions and gelation between proteins and increases the water-holding capacity [40], among other things, through the dissolution of collagen and the formation of gelatin [32]. Moreover, Kaur et al. [41] reported that cooking SV stimulates the activation of endogenous enzymes (cathepsins and calpains) at temperatures between 50 and 70 °C, but above 70 °C, the enzymes are completely inactivated [36].

It is well known that sensory characteristics are important criteria for consumer perception and acceptance [42]. The main benefit of using the SV technology is the optimal maintenance of quality without changing or improving the sensory properties of the food, such as desired color characteristics [43,44,45]. It causes minimal loss of moisture, less oxidation of lipids and proteins and modification of volatile flavor compounds. It also maintains the bioavailability of amino acids [32,42,46,47,48]. 

The main problem with the mild SV cooking method is the microbiological risk resulting from non-compliance with the standards of this method, e.g., very low process temperature, no inactivation of pathogens, the possibility of growth of anaerobic microflora and the potential for damage to the packaging especially during storage sous-vide products [26,31]. The use of low heat treatment parameters and improper use of this method may contribute to the lack of inactivation of pathogenic organisms and their multiplication [26]. There are possibilities for growth during storage of microorganisms such as *Clostridium perfringens*, *Clostridium botulinum* type E, and *Bacillus cereus*, and facultative anaerobes such as *Listeria monocytogenes*, as well as *Escherichia coli*, and in the case of pork hazards are *Trichinella spiralis* and *Toxoplasma gondii*, which has been shown in some studies [49]. 

There are few studies showing the combination of marinating with the use of natural ingredients and the sous-vide method to extend the shelf life of various types of meat and improve its sensory qualities [50,51,52]. 

Previous studies on this topic, combining marinating in fermented dairy products (FDP) and SV, showed positive effects in terms of meat safety (microbial quality and fat oxidation processes) and physical parameters such as color and texture, measured instrumentally [47,51,53]. However, the previous studies did not show the changes that occur in the muscle protein profile under the influence of marinating in FDP and cooking SV in relation to sensory evaluation, as well as color and texture measured instrumental. The aim of this study was to evaluate the effect of marinating in kefir, yogurt and buttermilk and the effect of SV cooking on changes in the myofibrillar protein profile of pork *m. Longissimus thoracis et lumborum* in relation to its sensory quality.

## 2. Materials and Methods

### 2.1. Raw Material

The material for the study was the middle part of the m. Longissimus thoracis et lumborum of pigs of the Polish Large White (Wielka Biała Polska) breed with a live weight of 120–130 kg. Muscles with a mean pH value of 5.8 were collected from the left and right side of 12 chilled carcasses (*n* = 24 muscle), 24 h after slaughter, from a local slaughterhouse, obtained from the same breeder. The samples were visually inspected, and any remaining external fat and fascia (connective tissue) were physically removed. Each muscle (*n* = 24) was divided into four parts: control (*n* = 24), marinated for 3 and 6 days in kefir (*n* = 24), in buttermilk (*n* = 24), and in yogurt (*n* = 24). There were a total of 96 samples, including 24 non-marinated control groups, and 72 marinated in various ways (*n* = 36 marinated 3 days, *n* = 36 marinated 6 days). Half of the meat samples were sous-vide cooked at 60 °C (control *n* = 12, marinated 3 days *n* = 18, marinated 6 days *n* = 18) and half at 80 °C (control *n* = 12, marinated 3 days *n* = 18, marinated 6 days *n* = 18). 

Muscle after pretreatment was cut into slices 3 cm thick and about 200 g ± 0.01 weight (steaks, *n* = 240), and then it was marinated.

### 2.2. Marinating Procedure

Dairy-fermented products (FDP), kefir (KE), buttermilk (BM) and yogurt (YO), were purchased from the manufacturer (Mlekovita, Wysokie Mazowieckie, Poland) from the same part of production. KE contained 2% fat, 4.8% carbohydrates, 3.4% protein and 120 mg/100 g of calcium. The composition of BM was 1.5% fat, 4.8% carbohydrates, 3% protein and 116 mg/100 g of calcium. In turn, YO contained 3% fat, 5.3% carbohydrates, 3.9% protein and 170 mg/100 g of calcium. The active acidity in the products was determined with a CPC-501 pH meter equipped with a pH electrode ERH-111 (Elmetron, Zabrze, Poland). The pH of the products was KE 4.61, BM 4.60 and YO 4.58.

The samples were prepared in appropriate quantities to carry out physicochemical and sensory analysis in accordance with the scheme shown in Figure 1.

The marinating process consisted of dipping the slices of muscles in the marinades: kefir, buttermilk and yogurt, and marinating for 3 or 6 days. After dipping in FDP, each slice of meat was immediately individually placed in a foil bag (80 GR vacuum cooking bags, ORVED S.p.A, Musile di Piave, Italy). The control group (C) was meat not marinated in FDP. The ratio between meat and marinade was fixed at 1:10. The bags were vacuum-packed (VAC-20 DT, Edesa Industrial S.L., Barcelona, Spain) and stored in a refrigerator (+4 °C) for marinating.

### 2.3. Sous-Vide Cooking

After 3 or 6 days of marination, the vacuum-packed samples were cooked in a gastronomic circulator, half of the samples at 60 °C and the other at 80 °C for 6 h. Sous-vide cooking conditions were established in previous studies [47]. The meat samples were coded as follows: KE6, BM6, YO6 and C6 (meat cooked at 60 °C) and KE8, BM8, YO8 and C8 (meat cooked at 80 °C). 

### 2.4. Processing Loss

Processing loss (%) was calculated by measuring the differences in the weight of slices of raw meat before and after marination, cooked sous-vide, cooled, removed from the bag and dried on filter paper. Three measurements were made for each sample.

### 2.5. pH Value

The pH of the mixture (crushed sample: distilled water 1:10) was measured using a digital pH meter CPC-501 (Elmetron, Zabrze, Poland) equipped with a pH electrode (ERH-111, Elmetron, Zabrze, Poland) in accordance with ISO 2917:1999 [54]. Three measurements were made for each sample.

### 2.6. Profile of the Muscle Proteins

The profile of the muscle proteins SDS−PAGE was performed to determine the raw muscle protein profile and after heat treatment following the method of Przybylski et al. [55] using a Bio−Rad apparatus. A sample of 20 mg meat was homogenized at 13,500 rpm (IKA DI 25 Ultra Turrax, Königswinter, Germany) with 600 µL of 0.003 M phosphate buffer at pH 7 for 1 min. The concentration of the protein was determined following the Bradford procedure [56]. The resulting homogenate was aliquoted and frozen at −80 °C for further analysis. 

Proteins were dissolved 1/1 (*v*/*v*) in buffer containing Tris-HCl, pH 6.8; 8 M urea; 2 M thiourea; 0.375 M b-mercaptoethanol; 3% SDS; and 0.05% bromophenol blue. The mixture was then heated at 95 °C for 3 min. Then, 15 mg of the protein mixed with glycerol (for a 10% final concentration) were loaded into each well of a 12% acrylamide Tris-HCl separating gel with a 5% stacking gel. Electrophoresis separation was carried out at room temperature for 1 h at 100 V and then for 6 h at 150 V. The gels were stained with 0.5% Coomassie Brilliant Blue R-250. Image analysis and quantification were performed using GelScan v.1.5 software (Kucharczyk TE—Electrophoretic Techniques, Warsaw, Poland). Thermo Scientific protein markers 11–245 kDa (Thermo Scientific Part No. 26616, 26617, 26618, Page Ruler Prestained Protein) were used as references for molecular weight determination. Densitometric analysis was used to estimate the quantity of a protein fraction, expressed as a percentage part of a band in a lane.

The following proteins were selected to analyze the results: nebulin, titin, myosin heavy chains (MHC), calpains ⴜ and µ, actin, and myosin light chains (MLC3).

### 2.7. Sensory Quality

The sensory evaluation of the samples after heat treatment was carried out using the scaling method according to the 5-point scale [57]. There were four training sessions. In the two first sessions, the color, odor, flavor and texture descriptors of SV cooked pork were studied and identified. All attributes are described, and points are defined for each attribute. The next two sessions concerned sensory of the quality of meat marinated and cooked at two different temperatures. Each attribute’s scale had marked anchors, as presented in Table 2.

The sensory analysis panel consisted of 6 people with confirmed sensory sensitivity and at least 5 years of experience [58,59]. Every panelist was selected and trained in accordance with ISO standards [60]. The members were between the group of 26–55 years. They were employees of the Department of Animal Food Technology. For evaluation, after SV cooking, the samples were cooled to room temperature and cut into cuboid-shaped pieces (1 × 1 × 3 cm). Individual samples for evaluation were placed in plastic containers with a lid (intended for contact with food) and marked with a special code. The test was carried out in a sensory laboratory, meeting the requirements of the standard [61]. Panelists rinsed their mouths with water between samples.

### 2.8. Instrumental Color Measurement

Color parameters were measured in the CIE *L** (lightness), *a** (redness) and *b** (yellowness) system on a freshly cut surface of the sample by reflectance method, using a spherical spectrophotometer 8200 Series (X-Rite Inc., Grand Rapids, MI, USA), using an illuminant D65, 10° observer angle and a 12 mm viewing area. White and black standards were used to calibrate the spectrocolorimeter. For each sample, fifteen measurements were made at random places on its surface. In addition, the color changes that were affected by the marinating process were determined using the color difference coefficient (Δ*E*), which was calculated relative to a control sample from each day of marinating time (3 and 6 days) and sous-vide temperature. The following Equation was used:(1)$${∆E}^{}=\sqrt{{(∆L}^{*}{)}^{2}+{(∆a}^{*}{)}^{2}+{(∆b}^{*}{)}^{2}}\;$$

### 2.9. Texture Profile Analysis (TPA)

Texture profile analysis (TPA) was performed using a texture analyzer (TA-XT2i, Stable Micro Systems Ltd., Godalming, UK). Fifteen cylindrical cores (13 mm length × 10 mm diameter) per treatment were cut from the central portion of the samples. Measurement was performed at room temperature. Samples were compressed twice between two parallel plates to 50% of their original height, a time interval of 5 s between the two compression cycles, at a crosshead speed of 2 mm/s. A TA-25 with a 2-inch diameter stainless probe was used. The calculation of TPA values was obtained using force and time plots. Test TPA recorded the following attributes: hardness (N) is the peak force that occurs during the first compression; springiness is measured by the distance of the detected height during the second compression divided by the original compression distance (dimensionless); cohesiveness is the ratio of positive peak force area during the second compression to the peak force area during the first compression (dimensionless); adhesiveness (N·mm) is the area under the curve for the first negative peak.

### 2.10. Statistical Analysis

The statistical analysis of the results was performed using STATISTICA software (version 13.3 PL StatSoft Inc., Tulsa, OK, USA). The mean of the results, standard deviation (SD) and multi-way analysis of variance (MANOVA) with Tuckey’s Honest Significant difference (HSD) post hoc test were performed. A significance level of α = 0.05 was used. In addition, the effects of marinating type, the marinating time, the sous-vide temperature and their interaction with each other were tested using MANOVA for a significance level of *p* ≤ 0.05, *p* ≤ 0.01, and *p* ≤ 0.001.

## 3. Results and Discussion

### 3.1. Processing Loss and pH Value of Meat Changes after Marinating Meat

Processing losses determine the cooking yield and are closely linked to the sensory properties of meat products, particularly juiciness, tenderness and other important quality characteristics. Juiciness and tenderness are the most important sensory attributes influencing consumer satisfaction with cooked meat [62,63]. The study showed (Figure 2 and Table 3) a significant effect (*p* ≤ 0.001) of marinating in FDP on processing loss. After 3 days of marination and heating at 60 °C, the products treated with yogurt and kefir showed significantly lower processing losses compared to the appropriate control samples, whereas buttermilk-treated pork showed comparable results with the other groups. After 6 days of marinating and heating at the same temperature (60 °C), the products with the addition of FDP showed similar processing losses compared to the corresponding control samples. Sour marinades for marinating are suitable for improving meat’s technological and functional properties. Many authors [14,53,64,65,66,67,68] confirm the beneficial effect of sour marinades on water retention in meat, related to the swelling of myofibrillar proteins and their increased extractability, as well as an increase in ionic strength and a decrease in pH.

As expected, SV temperature significantly (*p* ≤ 0.001) affected processing loss (Figure 2 and Table 3), and our results align with other authors. Samples cooked at 80 °C, regardless of the type of marinade and marinating time, showed significantly higher processing losses compared to the corresponding samples cooked at 60 °C. A 20 °C increase in temperature resulted in a 14 percentage point increase in processing loss. Based on the available data, which mainly concerns poultry and beef [27,35,42,62,69,70,71,72,73,74,75], and only a few pork samples [28,32,51,76,77], it can be concluded that increased juice loss and decreasing yield of the SV process correlates with increasing cooking temperature. Chotigavin et al. [70] found that SV cooking time was less important, as the highest SV cooking losses occurred during the first 0.5 h. In addition, Oillic et al. [78] found that sample size did not affect the level of processing loss.

As a rule, yields and water content are higher in SV products than in other conventional cooking [79,80,81,82]. The lower losses in the SV method are mainly due to the lower process parameters and vacuum packing, which minimize shrinkage and juice loss [40,83].

In addition to losing weight when cooking meat, there is a leakage of liquid containing mainly water and nutrients, which are mainly water-soluble [32,84]. Due to SV products retaining most of the nutrients [33,85,86,87]. In addition, Dominguez-Hernandez et al. [63] found that SV minimizes the coagulation of myofibrillar proteins.

Changes in meat structure can explain changes in leakage (reduced yield). Reduced yield and higher cooking losses are associated with protein denaturation and shrinkage of myofibrillar and sarcoplasmic proteins, along with shrinkage and solubilization of connective tissue, all contributing to reduced water-holding capacity [34,88]. According to Sánchez del Pulgar et al. [32], moisture loss in cooked meat is due to three main factors. First, water is lost through evaporation as the temperature increases. Second, temperatures above 40 °C cause denaturation and shrinkage of myofibrillar proteins. Actin and myosin begin to denature at 40–50 °C [89], changing sarcomere length and transverse contraction [34]. Myofibrillar proteins retain most of the water contained in muscle [42]. However, denatured proteins have less ability to bind water [89]. More significant gaps exist between the fibers, and more contained soluble protein and fat are released [42,89]—the consequence is weight reduction [29,88]. Further heating, between 56 and 62 °C, leads to a shrinkage of the muscle fibers and collagen [88,90] and, thus, reduces the meat’s ability to hold and bind water [91,92]. Up to 60 °C, the muscle fibers shrink transversely and widen the gap between the fibers, but above 60 °C, the muscle fibers shrink longitudinally and cause more water loss [42,75]. 

In line with this, Hwang et al. [83] and Zielbauer et al. [77] found that the processing loss of pork loin SV was lower after cooking at 50 °C compared to higher temperatures (above 55 °C). In these cases, the losses remain at the same level during prolonged heating at the set temperature. Sánchez del Pulgar [32] explains this by forming a gel from the dissolved collagen, resulting in higher water content in the meat and lower cooking losses. Reduced cooking losses in SV-treated meat benefit the meat industry due to higher yields of final products [32,63,93].

In the present study, the acidity of raw pork marinated in FDP was lower (*p* < 0.001) than the pH of the control sample, independent of the marinating time (Figure 3 and Table 3). Natural acidic marinades (fermented milk drinks, fruit juices, wine) contain weak organic acids, determining their pH. The pH of the buttermilk used in the marinade process was 4.60, that of kefir was 4.61 and that of yogurt was 4.58. However, due to the very similar pH values of the FDPs, no effect of the type of marinade on the pH of the raw meat was found. Other studies have also shown that marinated meat’s acidity depends on the marinade’s pH [51,66,68,94,95,96,97,98]. At the same time, the pH of the raw meat under the influence of the marinade decreases during the first three days of marinating, after which it stabilizes [12,98].

The pH of the raw pork after marinating was significantly lower than that of the control sample. Other authors [51,99] have reported the lower pH of marinated and heat-treated samples, but we did not observe this for most of the samples. After 3 days of marination and 60 °C heating, there was no difference in the pH values. But, after 6 days of marinating and heating at 60 °C, we found that the addition of BM significantly increased the pH compared to the other marinades. At the same time, it was found that SV, regardless of marinating time and cooking temperature, increases the pH by approximately 0.2 units. This may be due to protein hydrolysis, which results in an increase in ammonium nitrogen levels [100]. This study seems to confirm this hypothesis. The type of fermented dairy product and heat treatment temperature influenced calpain ⴜ and µ degradation, increased protein fractions and muscle protein degradation during heat treatment (Figure 3 and Table 3). Alkalization of the environment, however, could not neutralize the low pH caused by the acidic marinades.

### 3.2. Profile of the Muscle Proteins after Marinating and SV Cooking

Figure 4 demonstrates the results of protein separation by molecular weight (SDS-PAGE electrophoresis) before and after heat treatment. The conducted SDS-PAGE focused on the analysis of 4 selected myofibrillar proteins. Based on the estimation by protein ladder of 24 samples (8 raw and 16 after heat treatment SV), selected four myofibrillar muscle proteins were analyzed. We supposed that the identified proteins based on molecular weight are nebulin, titin, myosin heavy chains (MHC), calpains ⴜ and µ, actin, and myosin light chains (MLC3).

Densitometric analysis was used to estimate the content of a particular protein fraction in each sample. Selected significant results of this analysis are presented in Table 4. 

Changes in the protein profile and protein degradation during aging, marinating and after heat treatment are valuable markers of meat quality, especially its tenderness [55,101,102,103,104]. Our study was focused on discussing the myofibrillar proteins (myosin and actin), calpains and myosin light chains (MLC3), which are degradation products of myosin heavy chains (Figure 5, Table 4 and Appendix A). The effect of marinating type and sous-vide temperature on changes in the case of all protein profiles was statistically significant (*p* ≤ 0.001). However, the marinating time effect was significant in the case of calpains ⴜ and µ (*p* ≤ 0.05), and a strong marinating time effect (*p* ≤ 0.001) was observed in nebulin, titin, heavy chains myosin and light chains myosin. The impact of the type of fermented dairy product and the temperature of heat treatment significantly affects the inactivation (autolization) of calpains ⴜ and µ. The increased proportion of protein fractions: nebulin, titin, heavy chains myosin, calpains ⴜ and µ and actin in marinating but uncooked samples may be due to the low pH of fermented dairy products which increases the activities of proteolytic enzymes. 

As suggested by other authors [101], the increase in the share of myosin light chains in each sample indicates protein degradation, especially the fraction of myosin heavy chains. This is related to the protein hydrolysis affected by exogenous and endogenous enzymes, which determine the breakdown of proteins into peptides and amino acids caused by biochemical processes occurring during the marinating process [105]—especially with the hydrolysis of the meat protein caused by the microbial enzymes [106,107] and the presence of an acidic environment that favors the activity of endogenous meat proteases [108]. Moreover, according to Bee et al. [109], the autolysis of µ-calpain may have a key role in the degradation of proteins of meat. Along with the degradation of calpains, a simultaneous decrease in the content of other proteins in the tissue is observed, especially cytoskeletal proteins (nebulin, titin, heavy chains myosin and actin), which is confirmed by the results of our research. Furthermore, the degree of degradation of these cytoskeletal proteins affects the tenderness of the meat, thus affecting its sensory quality [104,110,111,112].

For samples after heat treatment, a reduction in the content of selected protein in the percentage of the lane was observed. With an increase in the heat treatment temperature, a significant decrease in the content of individual protein fractions in the samples to sous-vide treatment can be observed (*p* ≤ 0.001), regardless of the marinating type. In addition, the increase in the heat treatment temperature from 60 °C to 80 °C causes a two-fold decrease in the content of heavy chains of myosin, actin and calpains ⴜ and µ (*p* ≤ 0.001). This proves the degradation of muscle proteins during heat treatment. The same results have been obtained by other researchers [104,113,114]. In our research, the use of yogurt and kefir for marinating meat in combination with a higher temperature of heat treatment of meat (80 °C) causes the greatest degradation of muscle proteins, regardless of the marinating time (*p* ≤ 0.001).

### 3.3. Effect of Fermented Dairy Products Marinating and Sous-Vide Cooking on Pork Sensory Quality

Preferred sensory properties of marinated meat products are important for consumers. They depend on the quality of the raw material but also on the type of marinade [115]. In this study, it was shown that the type of marinade, marinating time and SV cooking temperature had a significant effect on (*p* ≤ 0.001) sensory attributes (Figure 6, Appendix A).

The analysis of the main components of the profile evaluation of meat samples marinated and stored after packing in bags for 3 and 6 days (M3, M6) and cooked sous-vide at temperatures 60 °C and 80 °C showed that the variability of the samples was mainly attributed to mainly the first principal component (71.43% of the total variability) and concerned the varied intensity of juiciness, hardness and tenderness. The second main component was assigned a much lower percentage of general variability—15.20%—which indicates that the intensity of the following discriminants: sour flavor, salty flavor and cooked meat flavor did not differentiate the assessed samples.

The samples cooked SV at 60 °C showed similarities in sensory quality and were completely different from the samples cooked sous-vide at 80 °C, as evidenced by the position of the samples on opposite sides of the system along the first principal component (Figure 6).

Vectors mark particular discriminants, where the length of the vector denotes the degree to which a given discriminant differentiates the examined set of samples; parallelism (or a very small angle between two vectors) and their unidirectionality means that these discriminants are positively correlated with each other; the opposite location of the vectors indicates their high but negative correlation (Figure 6).

Panelists rated the best color and uniformity of meat marinated for 6 days in KE or BM and cooked SV at 80 °C than at 60 °C (Appendix A). The lower ratings of SV meat cooked at 60 °C are due to its pink color, which is considered by consumers to be undercooked, while other cooking methods result in browning of the meat due to Maillard reactions [30]. The degree of myoglobin denaturation has the greatest influence on the intensity of reddening of cooked meat. Myoglobin denaturation occurs in the temperature range of 55–80 °C [63]. However, not all forms of myoglobin are equally sensitive to denaturation [86]. Deoxymyoglobin, which is the dominant form in vacuum-packed meat, is more resistant to heat denaturation than oxymyoglobin and metmyoglobin [116,117]. Previous studies [72] also showed that meat cooked SV at 60 °C had a pinker color than meat cooked SV at 80 °C.

Sensory evaluation showed the influence (*p* ≤ 0.001) of marinade type, time and temperature of SV cooking temperature on the texture. Overall, the panelists gave the highest marks to the meat marinated for 3 days in BM and cooked SV at 60 °C. The beneficial effects of acid marinate on sensory-evaluated texture attributes have been reported by many authors [67,95,97,99,118]. The texture of cooked SV meat is determined by the intensity of heat treatment, which affects the water retention capacity of the meat and the denaturation of meat proteins [32,33,35,42,62,71,119]. 

Long cooking of SV meat at lower temperatures results in better sensory characteristics, including tenderness and juiciness of products [73,120,121,122], because it minimizes the degree of fiber shrinkage [63]. Cooking loss correlates with the objective parameters of tenderness because protein coagulation is accompanied by meat hardness [123]. 

Dominguez-Hernandez et al. [63] also found that the lower cooking temperature of SV results in more tender meat, and juiciness also increases as temperature decreases. There was no influence of marinating time on meat hardness (*p* > 0.05), defined as the force required to compress the sample using teeth. Panelists similarly rated meat samples stored for 3 and 6 days. Whereas the marinating time had a significant effect on the other texture attributes. Longer marinating time (6 days) and higher temperature (80 °C) of SV cooking significantly (*p* ≤ 0.001) resulted in a low assessment of the texture of the samples, especially tenderness and juiciness. Samples marinated in yogurt and kefir for 6 days and cooked SV at 60 °C or 80 °C were less tender and less juicy than samples marinated for 3 days. In addition, longer marinating of meat in FDP and higher SV cooking temperature resulted in greater palpability of meat residue after chewing (adhesiveness to teeth). The best results in this attribute were obtained in the control sample, and the sample was marinated in KE, stored for 3 days and cooked SV at 80 °C. Similar results were obtained by Park et al. [72]. A significant influence (*p* ≤ 0.001) of the type of marinade, marinating time and SV cooking temperature was found on the overall odor sensation and the intensity of the odor of cooked meat.

A significant influence (*p* ≤ 0.001) of the type of marinade, marinating time and SV cooking temperature was found on the overall odor sensation and the intensity of the odor of cooked meat. The panelists found the odor of the control sample and samples marinated for 3 days in YO and BM and cooked SV at 60 °C to be highly desirable (Appendix A). Extending the marinating time and using a higher cooking temperature significantly decreased (*p* ≤ 0.001) the overall odor rating. The odor of cooked meat was more intense in samples cooked SV at 80 °C than 60 °C in the order C > BM > YO > KE. Despite the use of acidic marinades containing, among others, lactic acid, the panelists did not detect a sour odor in any of the analyzed samples. On the other hand, longer marinating in FDP caused a significant (*p* ≤ 0.001) decrease in the intensity of the odor of cooked meat. The least odor of cooked meat was found in the control sample after 6 days of storage and in the cooked SV at 60 °C. Among the marinated samples, this odor was the weakest in the samples marinated for 3 days in YO and BM and cooked at 60 °C. A similar effect was obtained in the KE sample marinated for 6 days, regardless of the SV cooking temperature. Research by other authors [99,124] also confirm the beneficial effect of marinating meat in fermented milk drinks on the odor attractiveness of meat products. 

A significant effect (*p* ≤ 0.001) of the type of marinade, marinating time and SV cooking temperature on the intensity of the sour flavor of cooked meat and other non-specific (foreign) odors was found. Interestingly, some panelists sensed a salty taste in the control sample but not in the marinated samples (*p* ≤ 0.01). The sodium cation is responsible for the salty taste [125,126], which in raw pork meat is approx. 0.4 g per kg. Marinating the meat in FDP effectively masked the salty taste resulting from the presence of sodium ions in the meat. Salt and sour flavor mixtures symmetrically affect each other’s intensity with enhancement at low concentrations and suppression or no effect at high concentrations [127]. This is due to the fact that sodium and sour ionic tastes involve the movement of stimulus ions into taste cells (absorption, not adsorption) [128].

The meat contains a variety of flavor compounds that are developed or enhanced by heat treatment above 70 °C [29,63]. Cooked meat flavor also comes from the Maillard reaction [63] and from lipid oxidation [72]. Lee et al. [129] reported that meat cooked at a high temperature (180 °C) has a higher level of flavor compared to SV meat cooked at 60 or 70 °C due to the lack of volatile aromatic compounds from the Maillard reaction. In contrast, the degree of lipid oxidation negatively affects palatability and consumer acceptance due to an increase in unacceptable taste [130]. Many authors indicate that higher cooking temperatures inhibit lipid oxidation [32,51]. Park et al. [72] confirm that meat SV cooked at a lower temperature has a worse taste than meat cooked at a higher temperature.

A lower cooking temperature increases yield and moisture retention in the product, which, in turn, according to Chotigavin et al. [70], may reduce the concentration of flavoring substances.

In this study, it was shown that longer marinating time (6 days), especially in kefir, increased (*p* ≤ 0.001) the intensity of sour and other flavor perception. These flavors were slightly less intense in the sample marinated at the same time in BM. Three-day pickling in YO and BM and cooking in SV at 60 °C had no significant effect (*p* > 0.05) on the sour flavor sensation compared to the control sample.

Other authors [99] showed a beneficial effect of short-term marinating in buttermilk on the intensity and attractiveness of the flavor. However, the extension of the marinating time had an adverse effect on the intensity of the flavor (a bitter flavor appeared). Other authors [124] using other fermented beverages (whey and sour milk) found the appearance of a sour flavor in the meat product.

### 3.4. Effect of Fermented Dairy Products Marinating and Sous-Vide Cooking on Pork Color Parameters

Color is an essential indicator of the technological quality of meat and meat products. The color of marinated meat is related to the color of the meat before marinating and to the pH of the marinade [99]. According to Fernandez-López et al. [131] and Pérez-Alvarez and Fernández-López [132], it is mainly related to the za-value of a specific heme pigment, the pigments’ chemical state, and the light-scattering and absorbing properties. Changes in the color properties of heat-treated meat products can be attributed to denaturation and oxidation of myoglobin and Maillard reactions [86]. Color is also one of the indicators of oxidative changes assessed in meat and meat products, as it depends not only on the content of heme pigments but also on their oxidation-reduction changes. An important factor shaping the color of meat is the redox potential, which determines the redox status of the iron centrally located in the porphyrin ring of the myoglobin molecule [99].

All three factors analyzed, i.e., marinade type, marinating time and cooking temperature, influenced (*p* ≤ 0.05) the lightness values (*L**) of the meat. Marinating in YO and KE and cooking SV at 80 °C significantly (*p* ≤ 0.001) lightened the color of the samples (Table 5 and Appendix A), while a change to a darker color (reduction of the *L** parameter) was observed in the sample marinated in BM and in the control sample cooked at the same temperature, thus confirming the results of previous studies [53]. Interestingly, greater stability of the *L** parameter, regarding the effect of cooking temperature, was observed in the samples marinated in YO and KE. The control sample and the sample marinated in BM cooked at higher temperatures (80 °C) were significantly (*p* ≤ 0.001) darker than those cooked at 60 °C. In all samples cooked at 60 °C, the brightness was at a similar level. Some authors [67,124] argue that the lighter color (higher *L** parameter values) of the marinated meat is due to the lower pH value of the meat after marinating, which leads to increased light scattering. This study and previous research [47] found no effect of marinade type on the pH value of SV-cooked samples, either at 60 °C or 80 °C. Changes in the degree of red saturation indicate the processes involved in processing and marinating meat [53]. The intensity of redness of cooked meat is inversely proportional to the degree of myoglobin denaturation [133], as is yellowness. In line with this, samples cooked with SV at a higher temperature had significantly higher *a** (*p* ≤ 0.001) and *b** values, which is consistent with the results of other authors [32,34,42]. Some authors [117,134] argue that the more significant redness of SV-cooked meat is due to deoxymyoglobin being more resistant to heat denaturation than other myoglobin forms. The type of marinade was not found to affect (*p* > 0.05) *a** values at shorter marinating times. In contrast, increasing the marinade time, especially in YO, to 6 days resulted in a significant increase (*p* ≤ 0.001) in redness in the SV-cooked sample at 60 °C and a decrease in redness in all marinated and SV-cooked samples at 80 °C compared to the control. It confirms our previous study [47]. In addition, the effect of marinating meat in buttermilk and whey for a more extended time on the reduction of redness in the high-temperature (baked) samples was also noted by Augustyńska-Prejsnar et al. [99], who believe that the fermented dairy beverages used for marinating reduce myoglobin oxidation. It is probably due to the protective antioxidant effect of bioactive peptides formed during the hydrolysis of milk proteins [135]. Changes in redox potential due to the addition of reducing compounds affect the transformation of myoglobin, causing a change in color and the release of non-heme iron from the myoglobin molecule [47,51].

According to Tomasevic et al. research [136], a clear color difference between meat products is perceptible if Δ*E* > 3.5. When 2 < ∆*E* < 3.5, an inexperienced observer also perceives the color difference. If 1 < ∆*E* < 2, only the experienced observer perceives the difference, while when 0 < ∆*E* < 1, the observer perceives no difference. From the pairwise comparison between the control sample and the marinated and cooked SV samples, the effect of marinating time on the perception of the color of the samples can be seen. The most remarkable color change was found in samples marinated for six days in kefir and cooked SV at 80 °C. The total color change for most samples was between 1 and 2 days, where only an experienced observer could perceive the difference. It was found that the color of the samples marinated in YO and KE for three days and cooked SV at 60 °C did not differ from that of the control, meaning that an observer would not notice a color difference (Δ*E*).

### 3.5. Effect of Fermented Dairy Products Marinating and Sous-Vide Cooking on Pork Texture Parameters

The texture is one of the most important quality characteristics of meat and its products. It influences the acceptance of meat among consumers. Previous studies [47,53,94,97,99,124] have shown that the low pH of meat after marinating in acidic marinades has a positive effect on texture. The compression method of texture profile analysis (TPA) is often used for the instrumental assessment of meat texture, miming the conditions that food is exposed to throughout the chewing process [137]. Meat texture analysis is based on the measurement of deformation occurring during the compression of the sample, determining meat parameters such as hardness, cohesiveness, elasticity or adhesiveness. Figure 7 and Table 6 show the effect of marinating type, marinating time and sous-vide temperature on cooking loss and instrumental texture parameters of meat. The effect of these three factors on hardness was statistically significant (*p* ≤ 0.001). The effect of the type of marinating (*p* ≤ 0,05), and especially the SV cooking temperature (*p* ≤ 0.001), was also significant for the adhesiveness and cohesiveness of the samples. However, marinating time did not affect (*p* > 0.05) adhesiveness or SV cooking temperature on meat springiness. 

Samples cooked at the higher SV temperature had a non-significantly higher hardness. In contrast, marinating in YO and BM significantly reduced hardness compared to the control. Furthermore, longer marination in these marinades was also found to reduce hardness. It is in line with studies by other authors [32,42,67,97,99]. 

The reduction in hardness and, thus, the increase in tenderness under the influence of acidic marinades can be explained in physicochemical and enzymatic categories. On the one hand, this characteristic is correlated with increased water retention associated with swelling, increased extraction of myofibrillar proteins, or both. These changes are mainly due to decreased pH and increased ionic strength [138], which result in weakened electrostatic interactions between myofibrillar protein chains. On the other hand, storing meat at low temperatures results in further maturation [139], and the presence of calcium [140] and an acidic environment [1], and the presence of an acidic environment also favors the activity of endogenous proteases, calpains and cathepsins. Furthermore, biochemical processes involving exogenous proteases from lactic fermented beverages occur during marinating [105]. Furthermore, at 60 °C, the main proteolytic enzymes, calpains, are inactivated [141], while cathepsins continue to function, with only 50% of their proteolytic activity [142].

Most of the changes in meat toughness observed during cooking are also due to the heat treatment’s effect on the meat’s protein components [73]. The effect of heat treatment is the denaturation and solubilization of muscle protein components, including sarcoplasmic proteins, myofibrillar proteins (particularly myosin and actin) and connective tissue proteins [143]. Previous studies have also demonstrated the lack of effect of SV meat cooking time and temperature on elasticity and cohesiveness [37,72,134]. In this study, as in previous studies [47,53], we found no effect of marinating in FDP and SV cooking on springiness. As in our previous study, we found that the type of marinade had only a small effect on cohesiveness (*p* ≤ 0.05), with meat marinated for six days and cooked at 60 °C regardless of the type of marinade having lower cohesiveness than those marinated for 3 (*p* ≤ 0.001). Overall, samples cooked at 60 °C had higher (*p* ≤ 0.001) cohesiveness than those cooked at 80 °C. Roldan et al. [42] explained the reduction in cohesiveness at higher cooking temperatures by weakening the fiber structure. 

Chewing describes the product of three parameters: hardness, elasticity and cohesiveness. Therefore, chewing is clearly strongly correlated with hardness, which has been confirmed in our previous studies [47,53]. For this reason, we do not show chewing results in this study. On the other hand, we observed that the type of marinade (*p* ≤ 0.05) and the SV cooking temperature significantly affected adhesiveness SV (*p* ≤ 0.001). The sample marinated for three days in BM had the highest adhesiveness. In contrast, samples marinated in YO and KE required a longer marinade to achieve increased adhesion similar to the control (after three days of storage). We found greater adhesion in samples cooked at 60 °C than at 80 °C. Similar results were obtained by Sanchez del Pulgar et al. [32], who explained this with a more significant amount of water seeded on their surface (after slicing this surface to produce the small samples used for TPA) and with a higher degree of water retention by these samples. 

### 3.6. Correlation

Correlation between TPA parameters values and selected sensory attributes (hardness, tenderness and juiciness) and nebulin, titin, myosin heavy chains, calpains ⴜ and µ, actin, and myosin light chains (LC3) for pork m. *Longissimus thoracis et lumborum* after the sous-vide process was determined using Pearson’s linear correlation, and their significance was set at *p* ≤ 0.001 (Table 7). Moreover, the correlation analysis performed between TPA parameters values and selected sensory attributes (hardness, tenderness and juiciness) and nebulin, titin, myosin heavy chains, calpains ⴜ and µ, actin, and myosin light chains (LC3) for pork m. *Longissimus thoracis et lumborum* after the sous-vide process showed significant correlations (*p* ≤ 0.001). We found that lower levels of actin, as well as nebulin/titin and myosin (HC), were highly correlated with higher adhesiveness as measured by the TPA test of SV-treated meat (−0.64, −0.62, respectively, *p* ≤ 0.001). On the other hand, there was a very high positive correlation between nebulin/titin and myosin (HC) levels and meat cohesiveness (0.83, *p* ≤ 0.001), as well as with meat sensory attributes: hardness, tenderness and juiciness (0.85, 0.91 and 0.91, respectively, *p* ≤ 0.001). The level of calpains ⴜ and µ was strongly positively correlated with both tenderness [N] as measured by the TPA test (0.50, *p* ≤ 0.001) and tenderness as assessed by sensory evaluation (0.59, *p* ≤ 0.001), as well as the other sensory attributes: hardness (0.55, *p* ≤ 0.001) and juiciness (0.58; *p* ≤ 0.001). In addition, a very strong positive correlation was shown between juiciness and firmness and crispness in the sensory evaluation (0.98 and 0.94, respectively, *p* ≤ 0.001). The results confirm that changes in proteins have a significant impact on determining the textural parameters of meat, especially tenderness and juiciness, which is also reflected in sensory evaluation. Therefore, conducting heat treatment in a specific way that takes into account protein degradation (denaturation process) seems to be an effective tool in developing cooked pork quality.

## 4. Conclusions

Changes in the protein profile and protein degradation during maturation, marinating and after heat treatment are valuable indicators of meat quality, especially tenderness.In the study, it was shown that the type of marinade, marinating time and SV cooking temperature had a significant effect on protein profile changes and sensory evaluation. Marinating for several days in FDP increased the proportion of some fractions of cytoskeletal proteins (nebulin, titin), myofibrillar proteins (actin and myosin heavy chains) and enzymatic proteins (calpains) in raw meat. The high degree of degradation of cytoskeletal proteins after heat treatment affects the tenderness of the meat. Marinating in kefir and yogurt and cooking at a higher SV temperature (80 °C) resulted in more significant protein degradation.The panelists rated better (higher) the taste, aroma, tenderness and juiciness of meat marinated in buttermilk and samples marinated for three days in FDP and cooked SV at 60 °C. PCA showed that the variability of the samples was mainly attributed to the first principal component (71.43% of the overall variability) and related to different intensities of juiciness, hardness and tenderness. The second principal component was attributed to a much lower percentage of the overall variability, 15.20%, indicating that the intensity of the aroma characteristics did not differentiate the samples evaluated. Samples of various marinated and SV cooked at 60 °C showed similarities in sensory quality and were quite different from samples of SV cooked at 80 °C.This study showed that marinating in dairy fermented products and SV cooking at 60 and 80 °C, taking into account the degree of protein degradation, can be an effective tool for shaping the sensory quality of cooked pork without the use of synthetic additives.

## Figures and Tables

**Figure 1 foods-12-03257-f001:**
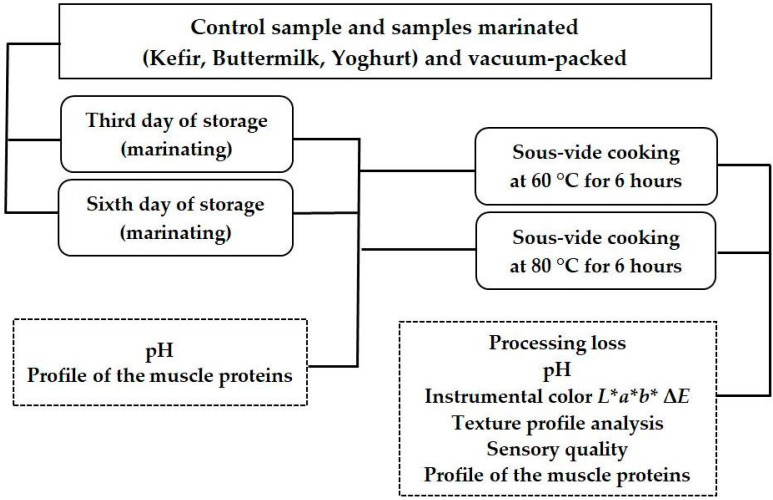
Schematic diagram of research.

**Figure 2 foods-12-03257-f002:**
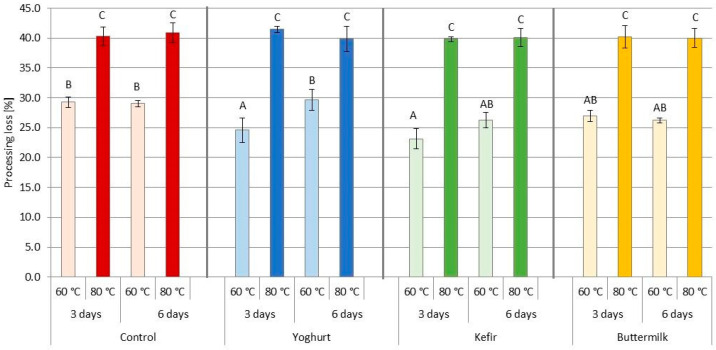
Effect of marinating type, marinating time and sous-vide temperature on pork *m. Longissimus thoracis et lumborum* processing loss (%); (A, B, …)—different letters indicate significant differences between means for samples marinated in liquid fermented dairy products (FDP) and control sample (HSD test: *p* ≤ 0.05). Red color of columns—Control samples, Blue color of columns—Yoghurt marinade samples, Green color of columns—Kefir marinade samples, Orange color of columns—Buttermilk marinade samples.

**Figure 3 foods-12-03257-f003:**
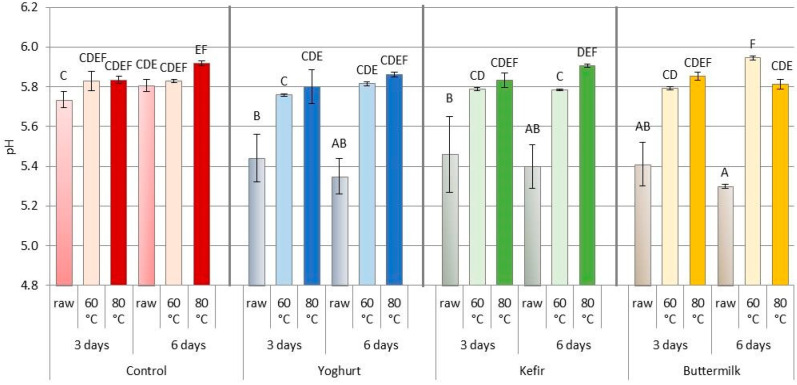
Effect of marinating type, marinating time and sous-vide temperature on pH values of pork *m. Longissimus thoracis et lumborum*; (A, B, …)—different letters indicate significant differences between means for samples marinated in liquid fermented dairy products (FDP) and control sample (HSD test: *p* ≤ 0.05). Red color of columns—Control samples, Blue color of columns—Yoghurt marinade samples, Green color of columns—Kefir marinade samples, Orange color of columns—Buttermilk marinade samples.

**Figure 4 foods-12-03257-f004:**
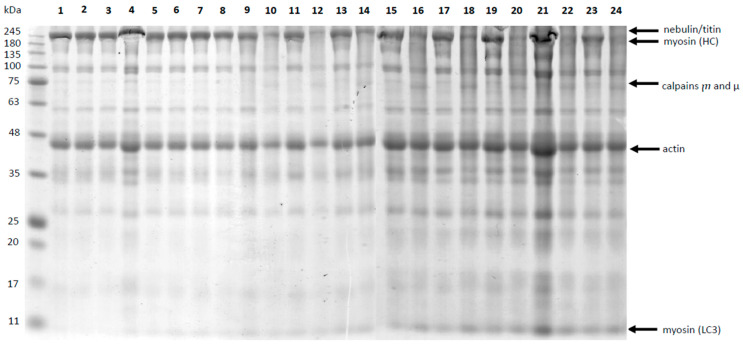
The results of protein separation by molecular weight (SDS-PAGE electrophoresis) before and after heat treatment; Raw: 1—Control_3 days_raw; 2—Yogurt_3 days_ raw; 3—Kefir_3 days_ raw; 4—Buttermilk_3 days_ raw; 5—Control_6 days_ raw; 6—Yogurt_6 days_ raw; 7—Kefir_6 days_ raw; 8—Buttermilk_3 days_ raw; After heat treatment: 9—Control_3 days_60 °C; 10—Control_3 days_ 80 °C; 11—Yogurt_3 days_60 °C; 12—Yogurt_3 days_80 °C; 13—Kefir_3 days_60 °C; 14—Kefir_3 days_80 °C; 15—Buttermilk_3 days_60 °C; 16—Buttermilk_3 days_80 °C; 17—Control_6 days_60 °C; 18—Control_6 days_ 80 °C; 19—Yogurt_6 days_60 °C; 20—Yogurt_6 days_80 °C; 21—Kefir_6 days_60 °C; 22—Kefir_6 days_80 °C; 23—Buttermilk_6 days_60 °C; 24—Buttermilk_6 days_80 °C; MHC—heavy chains; LC3—light chains.

**Figure 5 foods-12-03257-f005:**
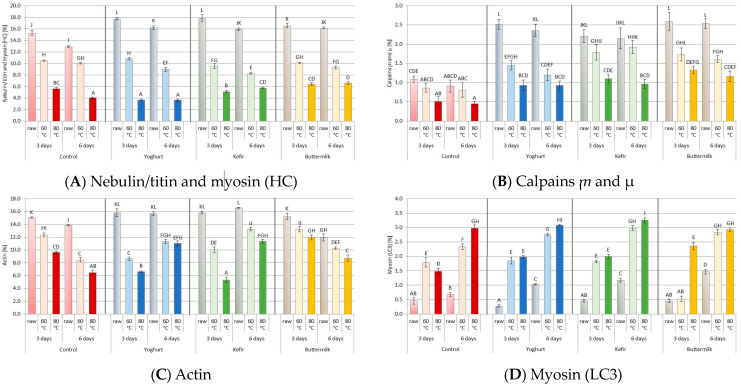
The results of quantitative densitometric analysis after SDS-PAGE electrophoresis (percentage of the band in the lane) for changes in (**A**). Nebulin/titin and myosin (HC); (**B**). Calpains ⴜ and µ; (**C**). Actin; (**D**). Myosin light chain (LC3) presents in pork *m. Longissimus thoracis et lumborum.* (A, B, …)—different letters indicate significant differences between means for samples marinated in liquid fermented dairy products (FDP) and control sample (HSD test: *p* ≤ 0.05). Red color of columns—Control samples, Blue color of columns—Yoghurt marinade samples, Green color of columns—Kefir marinade samples, Orange color of columns—Buttermilk marinade samples.

**Figure 6 foods-12-03257-f006:**
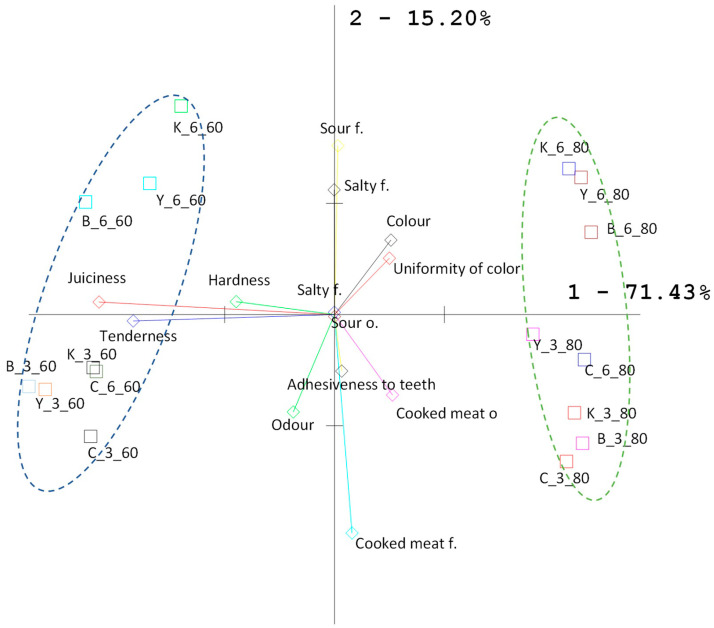
Graphical PCA projection of the results of sensory analysis. After heat treatment: Control—C_3_60 (Control_3 days_60 °C); C_3_80 (Control_3 days_ 80 °C); C_6_60 (Control_6 days_60 °C), C_6_80 (Control_6 days_80 °C); Yogurt—Y_3_60, (Yogurt_3 days_60 °C); Y_3_80 (Yogurt_3 days_80 °C), Y_6_60, (Yogurt_6 days_60 °C); Y_6_80 (Yogurt_6 days_80 °C), Kefir—K_3_60 (Kefir_3 days_60 °C), K_3_80 (Kefir_3 days_80 °C), K_6_60 (Kefir_6 days_60 °C), K_6_80 (Kefir_6 days_80 °C), Buttermilk- B_3_60 (Buttermilk_3 days_60 °C), B_3_80 (Buttermilk_3 days_80 °C), B_6_60 (Buttermilk_6 days_60 °C), B_6_80 (Buttermilk_6 days_80 °C); f.—flavor o.—odor.

**Figure 7 foods-12-03257-f007:**
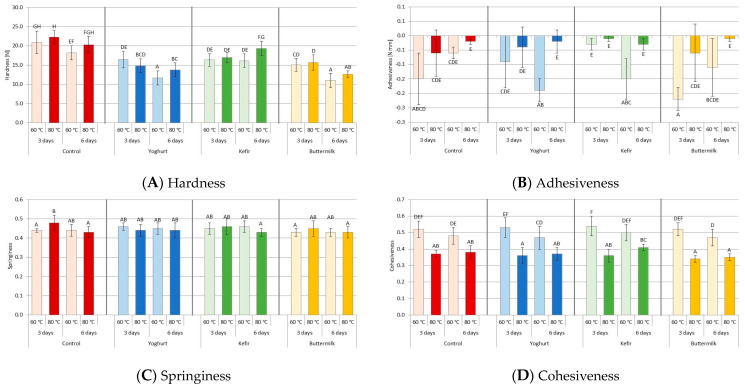
Effect of marinating type, marinating time and sous-vide temperature on instrumental texture parameters of pork *m. Longissimus thoracis et lumborum*. (A, B, …)—different letters in column indicate significant differences between means for samples marinated in liquid fermented dairy products (FDP) and control samples (HSD test: *p* ≤ 0.05). Red color of columns—Control samples, Blue color of columns—Yoghurt marinade samples, Green color of columns—Kefir marinade samples, Orange color of columns—Buttermilk marinade samples.

**Table 1 foods-12-03257-t001:** Natural marinades for meat.

Type of Marinade	Authors
marinade based on soy sauce and hot pepper paste	[4]
marinade with soy sauce, white wine, pepper, sugar, spices	[5]
marinade with wine and natural yogurt	[6]
marinade with wine or beer and olive oil	[7]
marinade using lytic bacteriophage and lactic acid	[8]
marinade with oregano and liquid smoke	[9]
marinade with rapeseed oil, spices and flavorings, and salt	[10]
marinade with red wine and salt	[11]
marinade with onion and garlic	[12,13]
marinade using potato tuber juice as a natural substrate for fermentation	[14]
marinade based on lemon juice	[7]
marinade with black currant juice	[15]
marinade with sour cherry and plum juice	[16]
marinade based on yellow mombin pulp mixed with water	[17]
marinade using Koruk (*V. vinifera* L.) juice and pomace with water	[18]

**Table 2 foods-12-03257-t002:** Sensory attributes, definition and scale anchors (scale 1–5).

Sensory Attribute and Definition	The Marks of Anchors
Color—perceived color tone	light pink (1), pink (2), light beige (3), beige (4), dark beige (5)
Uniformity of color—on the cross-section of the sample	no uniformity (1), slight uniformity (2), uniformity (3), average uniformity (4), high uniformity (5)
Total odor	unpleasant (1), quite unpleasant (2), neutral (3), quite pleasant (4), pleasant (5)
Intensity of sour odor—association with dairy-fermented products	none (1), light perceptible (2), clearly perceptible (3), intense (4), very intense (5)
Intensity of cooked meat odor	none (1), light perceptible (2), clearly perceptible (3), intense (4), very intense (5)
Intensity of sour flavor—association with dairy-fermented products	none (1), light perceptible (2), clearly perceptible (3), intense (4), very intense (5)
Intensity of salty flavor	none (1), light perceptible (2), clearly perceptible (3)—intense (4), very intense (5)
Intensity of cooked meat flavor	none (1), light perceptible (2), clearly perceptible (3), intense (4), very intense (5)
Intensity of other flavor—undefined flavor, e.g., off-flavor	none (1), light perceptible (2), clearly perceptible (3), intense (4), very intense (5)
Hardness—the force required to compress the sample using teeth	hard (1), light hard (2), neither soft nor hard (3), quite soft (4), very soft (5)
Tenderness—the effort required to chew the sample until it can be swallowed	big effort (1), medium effort (2), small effort (3), tender (4), very tender (5)
Juiciness—the amount of water in the sample released during 5 chews	dry (1), light dry (2), light juicy (3), juicy (4), very juicy (5)
Adhesiveness to teeth—the extent to which a product sticks to the teeth after chewing	much sticky (1), quite sticky (2), sticky (3), slightly sticky (4), no sticky (5)

**Table 3 foods-12-03257-t003:** Effect of marinating type, marinating time and sous-vide temperature on processing loss and pH values of pork *m. Longissimus thoracis et lumborum*.

Effect of:	Processing Loss [%]	pH Value
Marinating type	***	***
Marinating time	NS	NS
Sous-vide temperature	***	***
Marinating type × Marinating time	NS	NS
Marinating type × SV temperature	*	***
Marinating time × SV temperature	*	***
Marinating type × Marinating time × SV temperature	**	***

* = *p* ≤ 0.05; ** = *p* ≤ 0.01; *** = *p* ≤ 0.001; NS—non-significant effect = *p* > 0.05.

**Table 4 foods-12-03257-t004:** Effects of marinating type, marinating time and sous-vide temperature and its interactions on Nebulin/titin and myosin (HC), calpains ⴜ and µ, actin and myosin light chain (LC3) present in pork *m. Longissimus thoracis et lumborum*.

Effect of:	Nebulin/Titin and Myosin (HC)	Calpains ⴜ and µ	Actin	Myosin (LC3)
Marinating type	***	***	***	***
Marinating time	***	*	NS	***
Sous-vide temperature	***	***	***	***
Marinating type × Marinating time	***	NS	***	***
Marinating type × SV temperature	***	***	***	***
Marinating time × SV temperature	***	NS	***	***
Marinating type × and time × SV temperature	***	NS	***	***

* = *p* ≤ 0.05; *** = *p* ≤ 0.001; NS—non-significant effect.

**Table 5 foods-12-03257-t005:** Changes in instrumental color parameters in pork *m. Longissimus thoracis et lumborum* affected by marinating type, marinating time and sous-vide temperature.

Marinating		Sous-Vide Temp. [°C]	*L**	*a**	*b**	Δ*E*
Type	Time [Days]					
Control	3	60	81.3 ± 0.72 ^E^	1.7 ± 0.26 ^CDE^	11.6 ± 0.54 ^BCD^	-
		80	78.5 ± 0.92 ^AB^	2.8 ± 0.26 ^IJ^	13.2 ± 0.43 ^H^	-
	6	60	81.4 ± 1.37 ^E^	1.1 ± 0.40 ^A^	11.3 ± 0.38 ^ABC^	-
		80	78.1 ± 0.83 ^AB^	2.5 ± 0.42 ^HIJ^	12.5 ± 0.34 ^G^	-
Yogurt	3	60	81.0 ± 0.92 ^DE^	1.6 ± 0.23 ^BCD^	11.9 ± 0.25 ^DEF^	1.0 ± 0.18 ^AB^
		80	79.0 ± 0.29 ^BC^	2.9 ± 0.15 ^J^	13.4 ± 0.42 ^H^	0.5 ± 0.02 ^A^
	6	60	79.5 ± 1.91 ^CD^	1.8 ± 0.74 ^DEFG^	11.8 ±0.51 ^CDEF^	2.02 ± 0.24 ^B^
		80	79.8 ± 0.70 ^CD^	2.3 ± 0.31 ^FGH^	12.3 ± 0.31 ^FG^	1.33 ± 0.63 ^AB^
Kefir	3	60	79.8 ± 1.74 ^CD^	1.5 ± 0.34 ^ABCD^	11.6 ± 0.41 ^BCD^	1.5 ± 0.13 ^AB^
		80	78.9 ± 1.43 ^BC^	2.4 ± 0.38 ^HI^	13.2 ± 0.62 ^H^	0.4 ± 0.25 ^A^
	6	60	80.8 ± 1.29 ^DE^	1.3 ± 0.42 ^ABC^	11.7 ± 0.32 ^BCDE^	1.11 ± 0.11 ^AB^
		80	80.7 ± 0.83 ^DE^	2.1 ± 0.36 ^EFGH^	11.6 ± 0.33 ^BCDE^	3.50 ± 0.93 ^AB^
Buttermilk	3	60	80.6 ± 1.12 ^DE^	1.8 ± 0.38 ^CDEF^	11.2 ± 0.42 ^AB^	1.0 ± 0.14 ^AB^
		80	77.4 ± 0.76 ^A^	2.8 ± 0.27 ^IJ^	13.3 ± 0.31 ^H^	1.1 ± 0.48 ^AB^
	6	60	81.5 ± 0.96 ^E^	1.2 ± 0.36 ^AB^	11.0 ± 0.37 ^A^	1.54 ± 0.22 ^AB^
		80	78.5 ± 1.29 ^AB^	2.3 ± 0.34 ^GH^	12.1 ± 0.45 ^EFG^	1.15 ± 0.67 ^AB^
		SEM	1	0.14	0.17	0.18
Effects of:						
Marinating type			*	***	***	*
Marinating time			**	***	***	***
SV temperature			***	***	***	NS
Marinating type × Marinating time			***	*	NS	*
Marinating type × SV temperature			***	**	***	**
Marinating time × SV temperature			*	*	***	**
Marinating type × Marinating time × SV temperature			*	**	***	***

Mean ± SD; Δ*E*—determines the marinating impact and is calculated relative to a control sample from each marinating time (3 and 6 days) and sous-vide temperature. (A, B, …)—different letters in column indicate significant differences between means for samples marinated in liquid fermented dairy products (FDP) and control sample (HSD test: *p* ≤ 0.05); * = *p* ≤ 0.05; ** = *p* ≤ 0.01; *** = *p* ≤ 0.001; NS—non-significant effect = *p* > 0.05.

**Table 6 foods-12-03257-t006:** Effect of marinating type, marinating time and sous-vide temperature on texture parameters of pork *m. Longissimus thoracis et lumborum*.

Effect of:	Hardness [N]	Adhesiveness [N mm]	Springiness	Cohesiveness
Marinating type	***	*	*	*
Marinating time	***	NS	**	***
Sous-vide temperature	***	***	NS	***
Marinating type × Marinating time	***	***	NS	NS
Marinating type × SV temperature	*	NS	NS	NS
Marinating time × SV temperature	***	NS	***	***
Marinating type × and time × SV temperature	NS	**	*	NS

* = *p* ≤ 0.05; ** = *p* ≤ 0.01; *** = *p* ≤ 0.001; NS—non-significant effect = *p* > 0.05.

**Table 7 foods-12-03257-t007:** Correlation between TPA parameters values and selected sensory attributes (hardness, tenderness and juiciness) and nebulin, titin, myosin heavy chains (HC), calpains ⴜ and µ, actin, and myosin light chains (LC3) for pork *m. Longissimus thoracis et lumborum* after sous-vide process.

		TPA Parameters				Sensory Attributes			Proteins			
		Tenderness [N]	Adhesiveness [N mm]	Springiness	Cohesiveness	Hardness	Tenderness	Juiciness	Nebulin/Titin and Myosin (HC)	Calpains ⴜ and µ	Actin	Myosin (LC3)
TPA parameters	Tenderness [N]		0.19	0.19	0.01	−0.13	−0.22	−0.13	−0.08	−0.50 ***	−0.12	−0.23
	Adhesiveness [N mm]			−0.04	−0.60 ***	−0.70 ***	−0.61 ***	−0.69 ***	−0.62 ***	−0.44	−0.64 ***	0.37
	Springiness [-]				−0.12	0.06	−0.13	0.00	-0.05	0.00	−0.08	−0.19
	Cohesiveness [-]					0.89 ***	0.89 ***	0.93 ***	0.86 ***	0.55 ***	0.43	−0.33
Sensory attributes	Hardness						0.88 ***	0.98 ***	0.85 ***	0.55 ***	0.39	−0.33
	Tenderness							0.94 ***	0.91 ***	0.59 ***	0.39	−0.35
	Juiciness								0.91 ***	0.58 ***	0.39	−0.35
Proteins	Nebulin /titin and myosin (HC)									0.54 ***	0.44	−0.37
	calpains ⴜ and µ										0.48 ***	−0.16
	actin											−0.06
	myosin LC3)											

*** *p* ≤ 0.001.

## Data Availability

Data are contained within the article. The data used to support the findings of this study can be made available by the corresponding author upon request.

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
