# Peer review of "Effect of Marinating in Dairy-Fermented Products and Sous-Vide Cooking on the Protein Profile and Sensory Quality of Pork Longissimus Muscle"

_foods, 2023, doi:10.3390/foods12173257_

Round 1
Reviewer 1 Report
Research work is good but needs some corrections.
Comment on Manuscripts
“ Effect of marinating in dairy-fermented beverages and 2 sous-vide cooking on the protein profile and sensory quality of 3 pork Longissimus muscle”
1. Abstract is good but more findings of the research paper can be included and General statement. “Sensory attributes of meat products determine the willingness to buy and consume them. They can be shaped using natural acidic marinades and appropriate cooking methods.” should be deleted
2. . Introduction is written nicely but lack the proper justification of present work, where is methodology adopted with reference to other studied or not
3. In the manuscript material and method section is heavily crowded, author should have written it separately for each parameters.Specilly 2.1
4. Section 2.5 Sensory Quality should be given in detail? What of type of panel? Which of panel was used on which scale the test was carried? Why only 6 people if it is Trained panalist it should be 10 at least? Needs reference add below reference.
Utilization of lima bean starch as an edible coating base material for sapota fruit shelf-life enhancement. Journal of Agriculture and Food Research, 12,1-9
5. In the Colour analysis 2.6 for the sample Hue angle and Chroma was analysis ? Are they in the line of agreement with previous studied?.
6. Figure.1.Author has given several graphs, please explain the relevant of such complex graghs as these graphs are not in continues form and very difficult to understand from a point of view of readers.
7. Authors have given better figures for understands of viewers in various place but in few place it is bit not clear. I suggest change the figure in a simple form bar graphs.
8. Stastical analysis given by author is very good and neatly explained.
9.Section 2.7,2.8 and 3.1,3.4 3.5 and 3.6 are well explained and results are clear to understand
10. As per the results findings the Conclusion given is lengthe, it should be a complete gist of study in bullet points which is easy to understand for the the viewers.
Good
Reviewer 2 Report
Dear authors,
The research demonstrates extensive scope with promising results. It can be seen that an abstract is presented with very understandable information, as well as an introduction with comprehensive information. In addition, there are extensive discussions for most of the results.
However, the following observations should be reviewed in order for the manuscript to be considered for revision:
Abstract
Line 19: Please separate the symbol from the digit. Apply this throughout the manuscript where applicable.
Materials and methods
Lines 118-120: Please mention if the dairy products, kefir, buttermilk and yogurt were purchased in a supermarket (mention brand and country) or were made by the authors.
Results and discussion
It is suggested briefly discussing the results of the significance of Table 2 or placing the table after the information that refers to the processing loss information. This is so that there is more order to the way the information is presented.
Line 230: In the following sentence “... in temperature resulted in a 14 p.p. increase”… what does p.p. mean?
Line 233: Check the reference.
Figure 3: Consider reducing the number of literals for a better understanding.
Line 299: Is the way (02) reference mentioned correct?
Line 326: As I understand it, myosin and actin are part of the major contractile proteins (Myofibrillar) while nebulin and titin are cytoskeletal proteins. Please correct this.
Line 327: myosin light? Are the authors referring to light meromyosin?
Table 3: Literals must be placed after the standard deviation or the standard error. Regarding the latter, please clarify whether the standard deviation or standard error is referred to after the symbol ± (it is not mentioned in the legend). Please consider this detail in all the tables where this error occurs. Also, please reduce the number of literals for a better understanding.
In which part of the manuscript are tables S1 and S2 placed? In Supplementary Materials? These tables are not displayed anywhere.
Line 492: Is the term ‘za-value’ correct?
Lines 557-558: It seems that the information is incomplete. Please review and if necessary correct.
Conclusions
The conclusions are unclear and not concise, please restructure this section. Also, it is suggested that according to the variables evaluated, it is mentioned which was the best treatment.
Minor editing of English language required
Reviewer 3 Report
The authors would like to show the effect of marinating and sous-vide cooking at different temperatures on pork meat's protein profile and sensory quality. The topic is in the scope of the journal and shows some interesting data, despite several minor flaws found throughout the article. I suggest for revision of the article.
· Title. The word muscle is not supposed to be italicized.
· Line 110 - Please generate the formula in the form of a table to make it easier for the reader.
· Line 166 - What sensory test did you conduct in the study? is it hedonic acceptance or another test? The design is not the hedonic test. Suggest revising the sentence.
· Line 230 - what is the meaning of 14 p.p. ? is it an abbreviation or Units?
· Line 233-234, 284-285. This kind of mistake not supposed to be found in the peer review step.
· Revise the title of the figure and table, it is too long.
· Table - Please put the statistical results after the standard deviation ( example: 10±0.001A,B). I suggest to remove the information of p-value in table, as it is already represented by statistical letters. P-value is better mentioned in discussion.
Reviewer 4 Report
General Comments: This is a well-written article – one of the best I have been asked to review in a long time. Thank you!
However, I do have some major concerns regarding missing information within the experimental design and the description of the methods. Disclaimer: I am not an expert in SDS-PAGE; to the best of my knowledge this was conducted accordingly, but other reviewers should be given more weight on this topic. The sensory analysis method is not customary (hedonic scale; usually unmarked scales are used) for descriptive testing, but I see no reason why it could not objectively measure sensory properties. The sensory panel size is smaller than recommended for descriptive testing, but could suffice. For future studies, please try for 8-12 assessors on a panel (ISO 13299). The analysis of color change is not clear and it appears that deltaE maybe calculated using the difference from the control and the treatment group? This will confound effects, especially if the slices/samples are not taken from the same LTL. Please revise your analysis and results of color change according to the recommendation below. Finally, please provide a diagram or description showing where the sample material came from for each parameter measured. Was each sample divided into smaller aliquots for pH, TPA, sensory analysis, SDS-PAGE, etc. It is unclear just how many samples were analysed per parameter. This needs to be addressed throughout the manuscript (e.g. listing n= per treatment within the results section tables and figures, should the sample size differ per parameter), along with the animal effect (i.e. how many samples originated from each carcass and where did those slices end up in which treatments?), prior to considering publication. This will likely require re-running statistical analysis. Once this has been explained and analysis re-run, I would be happy to review this submission again. The research sounds very promising and relevant.
Specific Comments:
Line 47: please reconsider the use of “perfect”; it is an exaggeration. A recommendation would be “Fermented dairy products…can also be used as marinades.”
Line 113: were samples collected from both the left and right side of each carcass? Or which side? How many LTL were extracted per carcass?
Line 116: how many slices were cut from each LTL? 8 would be the ideal number to account for animal effects throughout the study design, but I doubt a single LTL is long enough for that.
Line 126: were the samples cooked directly in the marinade or were they removed and then SV cooked in a new vacuum-bag?
Please include information around lines 129 - 135 as to how animal effects were taken into account/balanced in the design. Please explain how the slices (lines 116) were allocated to a treatment group.
Line 139: replace “mass” with “weight”. The slices will have gravity acting upon them and therefore their mass is unknown.
Line 140: “a sample”? Was not the entire slice re-weighed to calculate % loss? Please rephrase. Why were three measurements necessary? Weighing usually does not have high variability and therefore measurement replication is not usual.
Line 146: why were 9 measurements required? Was the later average used? Was temperature of the sample also taken into account while measuring pH – this is important. If not, perhaps this is the reason you required so many replications of measuring pH as the sample warmed…
Line 153: do you have a primary source for the Bradford procedure? Please cite it.
Line 162: replace “performer” with “performed”
Line 188: how was color change measured? Usually it would be the change overtime for a single sample. This would mean that deltaE is available for the control group, yet it is not listed. Please correct your analysis and results for color change so that you are measuring change for a sample over time (i.e. 3days or 6 days).
Line 191: DeltaL*, deltaa*, and deltab* can also be interesting from a color change perspective and how marinating effects color overtime.
Line 196: how many cores were drilled per sample, as well?
Line 209: a bracket is missing in from of “(version”
Line 210: as the 48 samples were only taken from 12 animals, a two-way ANOVA needs to be run to account for animal effects. Animal has a large influence on texture (and feed on protein quality). As not all samples come from different animals this analysis is necessary. Please re-run your analysis including animal (where the LTL slice originated from) and repetition (you said there were three) as random effects in your statistical testing. If for some reason you did not keep track of which slices originated from each animal, please report this in the methods and report this major limitation and the implications of this lack of oversight in the discussion. Then, re-run your statistics with repetition as a random effect. If the repetition is not significant, then please report this in the section Statistical Analysis and feel free to keep the results as they are.
Language is great! Just a couple of minor errors that could slip past a native-speaker. ;-)
Reviewer 5 Report
See attached files.
Effect of marinating in dairy-fermented beverages and sous-vide cooking on the protein profile and sensory quality of pork Longissimus muscle
Agnieszka Latoch, Małgorzata Moczkowska-Wyrwisz, Piotr Sałek, Ewa Czarniecka-Skubina
The manuscript is quite interesting. The topic seems to be not very new. However, the authors should check the spelling and grammatic, as it is sometimes wrong and/or difficult to follow. I tried to start, but this is not my task.
The introduction is quite easy to follow. However, the authors could shorten the part, as there are often repeats (e.g. in the SV parts) and some sentences are complicated (e.g. lines 35-45). Many parts are quite speculative and should be clarified. Please clarify especially what is the difference, for example, between synthetic phosphates or acids and "natural" ones. Both are chemically phosphates or acids. There is a discussion, for example, about the difference of synthetic nitrates and nitrates and replacing these components with nitrite or nitrate rich plants. Chemical is chemical.
I tried to read everything but stopped in between, as the presentation and discussion of the results is difficult to follow. The results are not properly presented with regard to the significant differences between the groups, especially if there are interaction effects. The discussion is partly very speculative and there are many repeats. My biggest concerns are with regard to the PAGE results which are also very problematic and not well described.
I think that the manuscript is not perfect now in the Food journal.

See attached files.
Round 2
Reviewer 2 Report
Dear authors,
Although the manuscript has been improved according to the suggestions made by the reviewers, the following should be reviewed:
Tables and figures. I understand the authors’ answer which mentions “in this experiment there were 16 study groups so if there were significant differences between them, there is no way to reduce the number of literals, with the p-value applied”. However, I consider it important that the number of literals should be reduced for a better understanding. There are ways of reducing them, as the information at a glance is often confusing with many literals when the results should be clear.
Line 581: In the new version, it can be observed that the word ‘za-value’ has not been corrected.
Lines 744-747: I suggest that the conclusions be more concise and that the information in lines 744-747 be removed.
Minor editing of English language required
Author Response
Dear Reviewer 2,
Thank you for re-reviewing our manuscript. Revisions were made according to comments of Reviewer. The text has been checked by proofreader. The answers and comments are attached below. All changes in the manuscript were marked using track changes option in Word.
We hope that the improved manuscript will find your acceptance for publication. Thank you for your patience and help.
Kind regards
Authors
Dear authors,
Although the manuscript has been improved according to the suggestions made by the reviewers, the following should be reviewed:
Tables and figures. I understand the authors’ answer which mentions “in this experiment there were 16 study groups so if there were significant differences between them, there is no way to reduce the number of literals, with the p-value applied”. However, I consider it important that the number of literals should be reduced for a better understanding. There are ways of reducing them, as the information at a glance is often confusing with many literals when the results should be clear.
In this experiment there were 16 study groups so if there were significant differences between them, there is no way to reduce the number of literals, with the p-value applied. To make the results easier to understand and clear, we have presented them in the form of graphs.
We have not changed Table 5. Changes in instrumental color parameters in pork m. Longissimus thoracis et lumborum affected by marinating type, marinating time and sous-vide temperature, because we would like to show changes in delta E, indicating the changes that occur during the marinating and SV cooking, and this is not possible in the figure.
Line 581: In the new version, it can be observed that the word ‘za-value’ has not been corrected.
We corrected it - should be ‘value’
Lines 744-747: I suggest that the conclusions be more concise and that the information in lines 744-747 be removed.
Thank you for your comments, we corrected Conclusions section again.
Reviewer 4 Report
Thank you for making the required adjustments and comments. I no longer have any major concerns regarding this research.
Author Response
Dear Reviewer 4,
Thank you for re-reviewing our manuscript and kind words about our manuscript. Thank you for your patience and help.
Kind regards
Authors
Thank you for making the required adjustments and comments. I no longer have any major concerns regarding this research.
Reviewer 5 Report
OK
Author Response
Dear Reviewer 5,
Thank you for re-reviewing our manuscript. Thank you for your patience and help.
Kind regards
Authors